# Is Panoramic Radiography Really a Key Examination before Chemo-Radiotherapy Treatment for Oropharyngeal Cancer?

Carlo Bosoni [1,*], Michele Pietragalla [2], Davide Maraghelli [2], Vieri Rastrelli [2], Luca Giovanni Locatello [3], Isacco Desideri [4], Veronica Giuntini [1], Lorenzo Franchi [1] and Cosimo Nardi [2]

[1] Department of Experimental and Clinical Medicine, Orthodontics, The University of Florence, Via del Ponte di Mezzo, 50127 Florence, Italy; veronica.giuntini@unifi.it (V.G.); lorenzo.franchi@unifi.it (L.F.)

[2] Department of Experimental and Clinical Biomedical Sciences, Radiodiagnostic Unit n. 2, University of Florence-Azienda Ospedaliero-Universitaria Careggi, Largo Brambilla 3, 50134 Florence, Italy; michele.pietragalla@unifi.it (M.P.); davidemaraghelli@gmail.com (D.M.); vieri.rastrelli@gmail.com (V.R.); cosimo.nardi@unifi.it (C.N.)

[3] Department of Otorhinolaryngology, Careggi University Hospital, Via Taddeo Alderotti, 50139 Florence, Italy; locatello.lucagiovanni@gmail.com

[4] Department of Experimental and Clinical Biomedical Sciences, Radiotherapy Unit, University of Florence-Azienda Ospedaliero-Universitaria Careggi, Viale Morgagni 85, 50134 Florence, Italy; isacco.desideri@unifi.it

* Correspondence: carlo.bosoni@unifi.it

**Abstract:** Aim: To evaluate the diagnostic accuracy of panoramic radiography (PAN) for the identification of infectious foci of the tooth and periradicular bone before definitive chemo-radiotherapy treatment for oropharyngeal cancer, using multislice spiral computed tomography (MSCT) imaging as the reference standard. Materials and methods: 50 patients with oropharyngeal cancer who had performed both pre-treatment MSCT and PAN were retrospectively evaluated. Pre-radiotherapy MSCT showed 65 deep caries, 37 root remnants, 143 stage III periodontal diseases, and 77 apical periodontitis, for a total of 322 infectious foci. The same number of healthy teeth (control group) was selected via MSCT to be analysed by PAN. Sensitivity, specificity, positive and negative predictive values, and diagnostic accuracy for PAN images with respect to MSCT imaging were examined. Results: PAN showed sensitivity, negative predictive value, and diagnostic accuracy of 100% for deep caries, root remnants, and stage III periodontal disease, whereas there were 46.8%, 64.7%, and 72.1% apical periodontitis respectively. Conclusions: PAN did not show great diagnostic accuracy in the assessment of apical periodontitis, and therefore maxillofacial MSCT carried out before chemo-radiotherapy treatment should always be examined to identify dental and jaw diseases. Deep caries, root remnants, and stage III periodontal disease were perfectly detected on PAN.

**Keywords:** oropharyngeal cancer; radiotherapy; apical periodontitis; panoramic radiography; diagnostic accuracy

## 1. Introduction

Oropharyngeal carcinoma (OPC) accounts for 2.5% of all cancers and 9% of head and neck cancers. The most frequent histotype is squamous cell carcinoma, whose incidence has noticeably increased in recent years, with 98,412 new cases worldwide in 2020 [1], because of high-risk human papillomavirus infection (HPV) that is currently responsible for more than 70% of the OPCs [2]. On the other hand, smoking tobacco and alcohol consumption have been widely identified as the major risk factors for non-HPV-associated OPC [3].

The first clinical sign is usually non-painful cervical adenopathy associated with a few other symptoms including sore throat, otalgia, dysphagia, and odynophagia [4].

External radiotherapy (RT) provides excellent local control rates of early-stage OPC and it can be performed as a definitive treatment modality in combination with chemotherapy or as adjuvant therapy after surgical resection, even in advanced stages of the dis-

ease [5]. RT has shown to be a clearly effective treatment, especially in HPV-positive OPC patients with a three-year survival greater than 80% [6].

Multislice spiral computed tomography (MSCT) is a crucial volumetric imaging technique to correctly assess the clinical TNM staging, which is different between the HPV-related and classically tobacco/alcohol-related HPV-negative OPC [7]. Pre-irradiation dental-periodontal screening is usually carried out by means of clinical assessment and a panoramic radiography (PAN) so as to examine oral mucosa, bone tissue, and tooth inflammations. The ultimate aim is to minimize complications both during or immediately after RT, e.g., mucositis, xerostomia, and oral mucosa keratinization, and in the long term, atrophy, masticatory muscle fibrosis, trismus, and osteoradionecrosis [8]. PAN plays a key role in the diagnosis of oral infectious foci taking advantage from its panoramic view, low radiation dose, and low cost [9]. Nevertheless, a volumetric imaging technique such as MSCT has better diagnostic accuracy than PAN in the identification of dentoalveolar structures and especially in the detection of periapical bone lesions [10].

Post-RT complications are notably reduced by extracting untreatable dental elements and teeth with an uncertain prognosis before the radiation treatment begins [11]. However, there are no guidelines that indicate what criteria should be used to determine whether a tooth must be extracted or fixed before a RT treatment [12], nor are there any papers on the diagnostic accuracy of each available imaging techniques in the identification of pre-existing dental alterations.

The aim of this retrospective study was to evaluate the diagnostic accuracy of PAN in the detection of deep caries, root remnants, stage III periodontal disease, and apical periodontitis (AP) in a cohort of OPC patients that will have to be treated by RT, using MSCT imaging as the reference standard.

## 2. Materials and Methods

### 2.1. Patients and Devices

From January 2010 to December 2020, all patients (n°154) with a histological diagnosis of oropharyngeal squamous cell carcinoma undergoing RT (with or without chemotherapy) at the Careggi University Hospital (Florence, Italy) were selected. Of the originally enrolled sample, 27 patients who had undergone surgery and/or chemotherapy without RT and 5 patients who had received palliation were excluded. A further 58 patients were also excluded due to the lack of PAN prior to the start of RT, since PAN had been performed in other institutions or more than 40 days before treatment. Then, 5 patients were excluded because they had performed pre-RT MSCT elsewhere. Finally, 2 and 7 patients were excluded due to the presence of motion and metal artefacts that significantly degraded the quality of pre-RT MSCT images (Figure 1).

Exclusion criteria:

- Underage patient (<18 years);
- No histological diagnosis of OPC;
- No RT;
- No pre-RT PAN examination;
- PAN has not been performed during the 40 days prior to MSCT examination;
- No pre-RT MSCT examination;
- Poor MSCT image quality due to motion and/or metal artefacts.

Inclusion criteria:

- Adult patient (≥18 years);
- Histological diagnosis of OPC;
- RT for OPC;
- Pre-RT MSCT examination;
- PAN carried out during the earlier 40 days to MSCT examination;
- Absence of motion and/or metal artefacts.

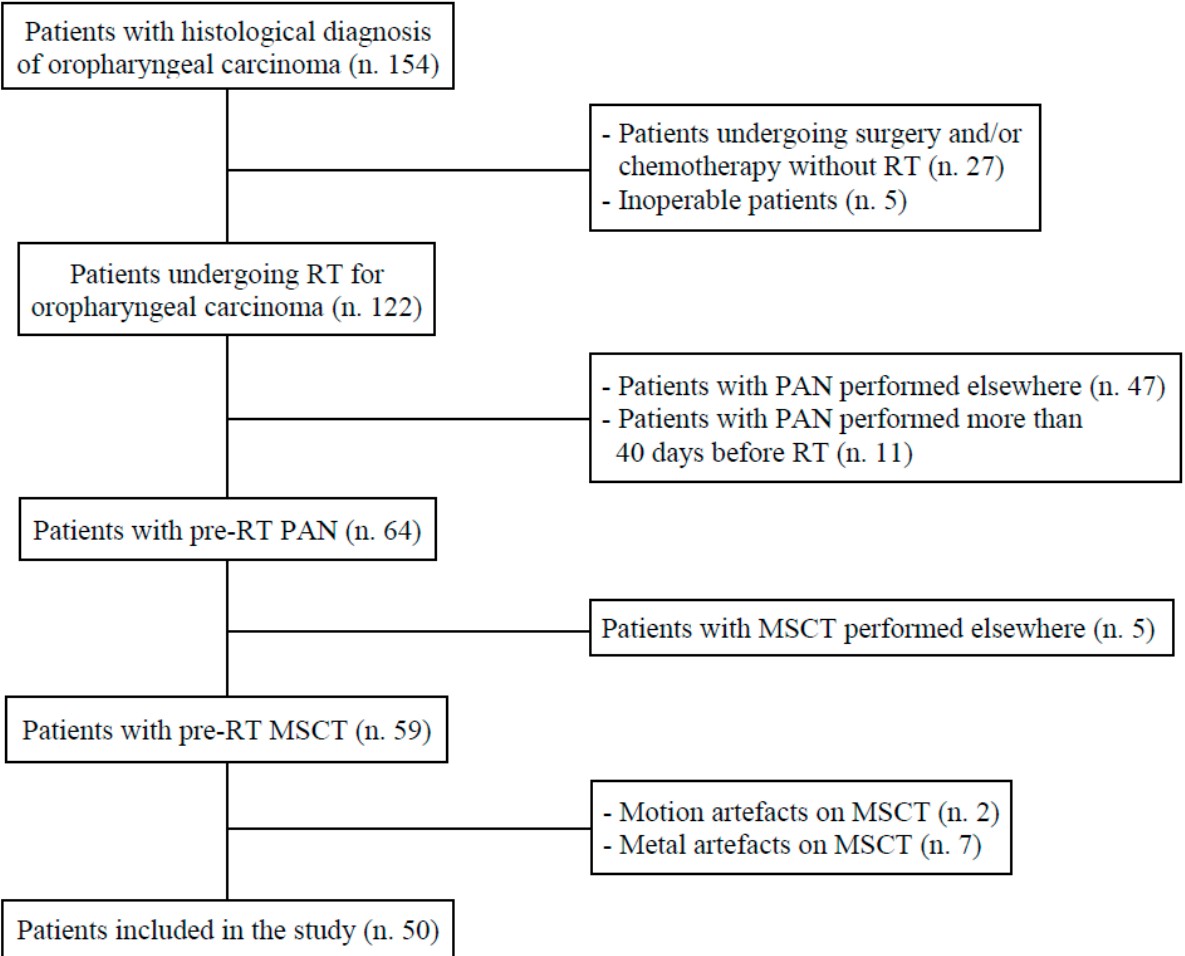

**Figure 1.** A flowchart of the selection criteria for enrolling patients. Radiotherapy (RT). Multislice Spiral Computed Tomography (MSCT). Panoramic radiography (PAN).

The final sample consisted of 50 patients with histologically confirmed squamous cell OPC (range 43–81 years; 37 males, mean age 62.3; 13 females, mean age 64.7) who carried out pre-RT MSCT and PAN during the 40 days prior to MSCT imaging.

All MSCT examinations were carried out with a 128-detector row helical CT scanner (SOMATOM Definition Flash, Siemens Healthcare, Erlangen, Germany) with a field of view extended from the skull base to the thoracic inlet. The following parameters were used: tube voltage 120 kV, current × exposure time 186 mAs, rotation time 0.5 s, pixel size 0.465 mm, both thickness and reconstruction intervals 1 mm, beam collimation 128 × 0.6 mm, and pitch 0.80. Post-processing, 1-mm-thick sections were obtained on axial, sagittal, and coronal planes oriented on the longitudinal axis of the tooth in question to simulate cross-section obtained by Dentascan software. The bone window was used for image evaluation.

PAN was performed via the Orthoceph OC200 D (Instrumentarium Dental, Tuusula, Finland). It was a digital panoramic radiograph with a rotation time of 17.6 s, 66 kV, and 4.2–7.7 mA.

MSCT and PAN images were displayed on a 20-inch medical monitor with a 3-megapixel Barco display (Barco, Kortrijk, Belgium) and 2048 × 1536 resolution. The software programs originally provided with the systems were used for image evaluation.

### 2.2. Assessment of Infectious Foci and Outline of the Study

For each tooth on MSCT, the presence or absence of four pathological conditions requiring treatment or extraction before RT was assessed [13]:

- Deep caries, defined as a loss of dental crown extending beyond half the thickness of the dentin with possible involvement of the pulp chamber [14];
- Root remnants, where there is loss of at least 75% of the crown [15];
- Stage III periodontal disease, determined by loss of bone extending to the middle third of the root or beyond [16];
- AP, defined as a periapical radiolucent area with changes in structures of bone that have been in contact with root apex and measured at least twice the width of the periodontal ligament space [17,18].

The presence of deep caries, root remnants, stage III periodontal disease, and AP was clinically/surgically confirmed by formal in-office evaluations of Florence University's dental clinic.

A total of 65 deep caries, 37 root remnants, 143 stage III periodontal diseases, and 77 AP lesions were detected by MSCT scans for a total of 322 infectious foci. Afterwards, 322 healthy teeth were selected as a control group (or healthy group without infectious foci) corresponding in terms of patient age and tooth type to those with infectious foci. When possible, the contralateral healthy tooth of the same patient was selected. Finally, the four pathological conditions of the 322 pathological teeth and 322 healthy teeth were analysed on PAN.

### 2.3. Readers and Statistical Analysis

Two head and neck radiologists recruited the 322 teeth by MSCT imaging for both diseased and healthy groups. Each tooth was independently assessed on PAN by a dental radiologist and a dentist skilled in PAN images. They were appointed over and above the two radiologists assigned to the choice of the 322 teeth and were blinded to any information about the patient/tooth selected. Whenever the MSCT or PAN readers came to different conclusions, a discussion was held until they reached a consensus.

Sensitivity, specificity, positive and negative predictive values, and diagnostic accuracy of PAN images were calculated in relation to the reference standard represented by MSCT. For each of the four parameters, the Cohen's K coefficient was calculated to assess the agreement between PAN and MSCT. In addition, Cohen's K coefficient was used to calculate inter-reader concordance for the categorical variable defined by the presence or absence of the four parameters at both PAN and post-RT MSCT. K-values of 0.01–0.20, 0.21–0.40, 0.41–0.60, 0.61–0.80, 0.81–0.99, and 1 corresponded to weak, poor, moderate, good, excellent, and perfect agreement, respectively. A *p*-value of $\leq 0.05$ was considered statistically significant. The collected data were analysed using the SPSS® v. 24.0 statistical analysis software (IBM Corp., New York, NY, USA).

### 3. Results

In the overall assessment of the four parameters, the K coefficient showed excellent inter-reader concordance (k = 0.86) in the analysis of PAN.

True positives, false positives, true negatives, false negatives, sensitivity, specificity, positive predictive value, negative predictive value, and diagnostic accuracy of PAN for each of the four parameters (deep caries, root remnants, stage III periodontal disease, and AP lesions) are shown in Tables 1 and 2.

**Table 1.** True Positives, False Positives, True Negatives, and False Negatives for Panoramic Radiography (PAN) in Relation to Multislice Spiral Computed Tomography (MSCT).

| PAN | Multi Slice CT | | |
|---|---|---|---|
| | **Diseased** | **Healthy** | **Total** |
| Deep caries | | | |
| Positive | 65 (100%) | 0 | 65 |
| Negative | 0 | 65 (100%) | 65 |
| Total | 65 | 65 | 130 |
| Root remnants | | | |
| Positive | 37 (100%) | 0 | 37 |
| Negative | 0 | 37 (100%) | 37 |
| Total | 37 | 37 | 74 |
| Stage III periodontal disease | | | |
| Positive | 143 (100%) | 0 | 143 |
| Negative | 0 | 143 (100%) | 143 |
| Total | 143 | 143 | 286 |
| Apical periodontitis | | | |
| Positive | 36 (46.8%) | 2 (2.6%) | 38 |
| Negative | 41 (53.2%) | 75 (97.4%) | 116 |
| Total | 77 | 77 | 154 |

**Table 2.** Sensitivity (SEN), specificity (SPE), positive predictive value (PPV), negative predictive value (NPV), diagnostic accuracy (ACC) and Kappa Value for Panoramic Radiography in Relation to Multislice Spiral Computed Tomography.

| Parameter | SEN | SPE | PPV | NPV | ACC | K |
|---|---|---|---|---|---|---|
| Deep caries | 100% | 100% | 100% | 100% | 100% | 0.96 |
| Root remnants | 100% | 100% | 100% | 100% | 100% | 0.94 |
| Stage III periodontal disease | 100% | 100% | 100% | 100% | 100% | 0.93 |
| Apical periodontitis | 46.8% | 97.4% | 94.7% | 64.7% | 72.1% | 0.45 |

No false positive or false negative was found in the assessment of deep caries, root remnants, and stage III periodontal disease, whereas false positives were very few (2.6%) (Figure 2) and false negatives accounted for just over half of the cases (53.2%) for AP lesions (Figure 3).

The agreement between PAN images in relation to the reference standard represented by MSCT was moderate in the analysis of AP (k = 0.45) and excellent in the analysis of caries, root remnants, and stage III periodontal disease (k = 0.93 to 0.96). PAN showed sensitivity, negative predictive value, and diagnostic accuracy of 100% for deep caries, root remnants, and stage III periodontal disease, whereas they were 46.8%, 64.7%, and 72.1% for AP lesions, respectively.

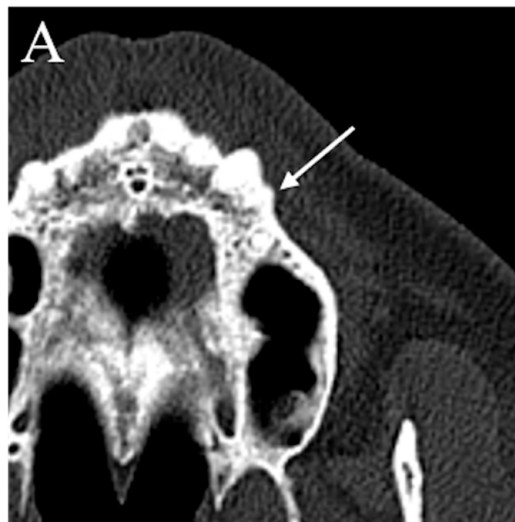 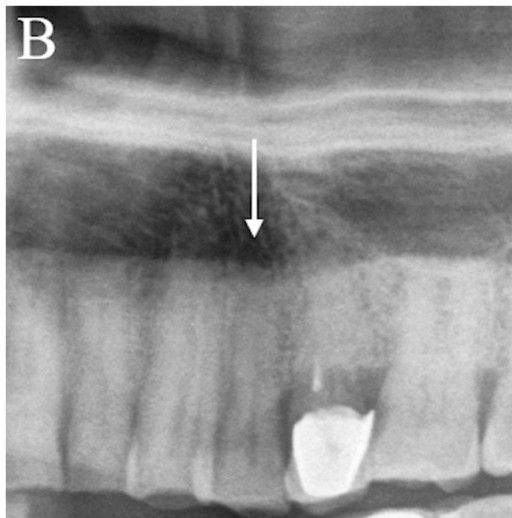

**Figure 2.** An example of false-positive apical periodontitis. (**A**) Upper jaw MSCT imaging. No periapical bone lesion was detected (arrow). (**B**) In PAN, at the level of the periapex of the root of the left first premolar, the observers noted a radiolucent periapical image (arrow) characterised by supposed changes in bone structure that simulated an apical periodontitis lesion.

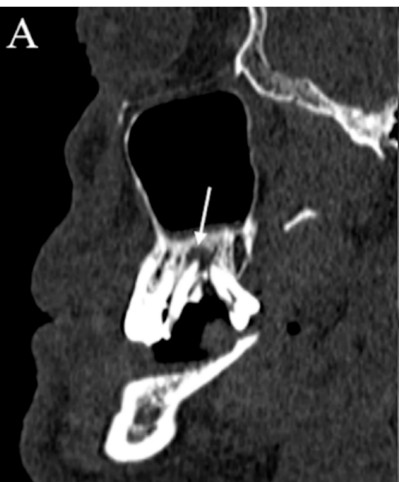 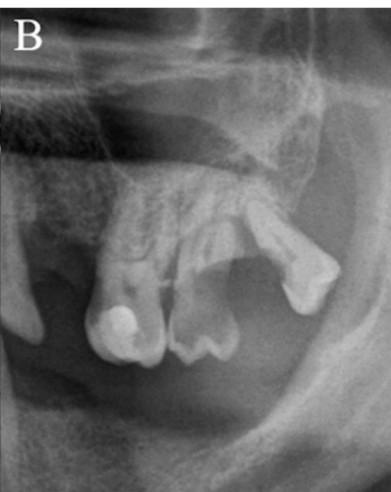

**Figure 3.** An example of false-negative apical periodontitis. (**B**) Upper jaw MSCT imaging showing apical periodontitis affecting the left second molar. (**A**) In PAN, no periapical bone lesion was detected at the level of the periapex of the left second molar.

## 4. Discussion

PAN showed excellent diagnostic accuracy in detecting deep caries, root remnants, and stage III periodontal disease, whereas it was not so accurate in the identification of AP lesions. In our opinion, the perfect accuracy of PAN in the evaluation of deep caries, root remnants, and stage III periodontal diseases was due to the advanced stage of the four pathological conditions that required remediation or extraction prior to radiotherapy. The loss of more than half of the dentine thickness in deep caries, loss of at least 75% of the crown in root remnants, and loss of bone to at least the middle third of the root in stage III periodontal diseases were all easily detected with PAN.

Generally, the accuracy of PAN for the identification of AP lesions depends on the anatomic area, cortical bone involvement, and lesion size [19]. The air within the maxillary sinus, the presence of numerous roots infrequently orthogonal to the X-ray beam, the undulating morphology of the maxillary sinus floor, and the anterior portion of the zygomatic arch superimposed on dentoalveolar structures are all possible obstacles to

the identification of AP lesions in the upper molar areas and to a lesser degree in upper premolar/canine areas. The evaluation of upper and lower incisive areas is also hard to carry out due to the morphologic diversities of chin/mental fossa and superimposition of the cervical spine, skull base, hard palate, and nasal bone/cartilage/air. On the contrary, AP lesions are well recognizable in lower premolar/canine and molar areas since the superimposition of extraoral anatomic structures is limited and roots are more orthogonal to the X-ray beam, although the projection of nerve canals and foramen may correspond to root apexes [9]. Furthermore, periapical radiolucency has to involve around 30–50% of the bone mineral loss to be radiographically recognizable [20]. Such value represents the needed threshold for the radiographic detection of AP lesions. The involvement of cortical bone facilitates the detection of AP lesions leading to earlier attainment of the bone demineralization threshold value since mineral content is greater in cortical than cancellous bones. For the same reasons, AP lesions involving a large amount of bone tissue are easier to identify than smaller ones [21,22].

Studies conducted by Estrela et al. [23] and Nardi et al. [9,24] showed that the accuracy of PAN in the identification of AP lesions was 53.8%, 65.0%, and 71.3%, respectively. In these studies, the reference standard for the assessment of AP lesions was cone beam computed tomography (CBCT). All teeth examined by Estrela et al. [23] had a history of endodontic infection confirmed by clinical examination, whereas the studies by Nardi et al. [9,24] investigated untreated and endodontically treated teeth with clinical/surgical diagnosis of the lesions. Another comparative study by Rios-Santos et al. [25] between PAN and periapical radiography indicated that AP lesions detected on PAN were 58.0%, although the authors did not mention how AP lesions had been confirmed. In the current study, the accuracy in diagnosing AP lesions on PAN was only 72.1%, thus confirming the undefined role of PAN in identifying AP lesions.

Our study assessed the accuracy of PAN in cancer patients, which is why the reference volumetric technique could only be MSCT. CBCT is not normally used in tumour staging because it is not appropriate for soft tissue assessment [26] and it is too slow (5–40 s) for the administration of intravenous contrast medium with non-negligible motion artefacts [27,28]. However, it has been widely demonstrated that both the study of bone quality in CBCT and MSCT are superimposable [22,29] and that MSCT accurately identifies caries, root remnant, and periodontal disease [30,31]. For this reason, we felt it was reasonable to compare the accuracy of two-dimensional PAN imaging with the volumetric reference standard, whether represented by MSCT as in the current study or by CBCT in previous papers [9,23,24].

The study of infectious foci in anticipation of RT treatment is of fundamental importance for the prevention of long-term complications. The well-known effects caused by RT are vascular alterations of both soft tissue and bone, damage to salivary glands, and increased collagen synthesis resulting in fibrosis [32,33]. Tooth extraction in an irradiated field is a risk factor for both infection and osteonecrosis, and remediations prior to RT are thus strongly recommended so as to remove mobile or decayed teeth, teeth with periapical lesions, and teeth causing trauma or mucosal lesions [34]. These procedures should be performed at least fourteen days before the beginning of RT to allow tissue healing [35].

In our opinion, a rapid diagnostic technique able to adequately assess the prognosis of each tooth and the bone tissue is of primary importance in this regard. Due consideration must be given to the best diagnostic technique to evaluate dentoalveolar structures, as PAN has proved to be excellent in the identification of deep caries, root remnants, and advanced periodontal disease, but poor in the detection of AP lesions. Patients with OPC are already staged in the diagnostic workup on the basis of a head and neck MSCT scan. Therefore, the same radiation oncologist might not only perform the clinical examination, but also evaluate the maxillofacial MSCT scan together with his/her radiologist colleagues in order to decide whether the patient needs a dental examination or has to undergo directly to RT. In this way, not only patients in an advanced stage of the disease, but also those with no

foci to be cleared could start therapy immediately, based on a shared assessment and with an obvious reduction of healthcare resources.

Future studies should explore the clinical effectiveness of such a swift protocol that would avoid PAN in this setting. In addition, subjecting the patient to additional ionizing radiations without providing diagnostic benefits might not be acceptable from a radiation protection point of view. Any exposure to radiations is justified only when patients can benefit from it [36] and basing dental-periodontal screening on PAN may lead to more long-term complications since pathological teeth may not be extracted or restored before RT begins. Nevertheless, clinical settings in which maxillofacial MSCT cannot be performed absolutely need PAN to assess dentoalveolar structures. These situations include claustrophobic people, those suffering renal failure, since these patients are commonly diverted to no contrast magnetic resonance imaging because of its intrinsic high contrast resolution, and MSCT protocols with a thickness and/or reconstruction interval greater than 1 mm, thus not acceptable to appropriately identify small alterations of bone tissue surrounding teeth.

Limitations of the study include the fact that we considered caries, root remnants, and periodontal alterations in advanced stages of disease and that we grouped all AP lesions regardless of their size, anatomical location, and bone cortical involvement. As has been previously demonstrated [9,24], there is much difference in the ability of PAN to detect AP lesions larger or smaller than 4.5 mm, located in the upper or lower arch, and causing resorptions of cortical bones. Another weakness was represented by the absence of histopathological examination of AP lesions because of the difficulty to perform it in routine clinical practice. What is not a real limitation to date, but that may become such in the future, is the more recent tendency to replace MSCT with magnetic resonance imaging in both tumour staging and follow-up. Magnetic resonance is an imaging technique with high contrast resolution and excellent representation of soft tissues, but with lower spatial resolution and less accuracy for the evaluation of hard tissues, such as teeth and bone, compared to MSCT [37,38].

## 5. Conclusions

In our series, PAN did not show satisfactory diagnostic accuracy in the assessment of AP lesions, whereas it proved to be excellent in the identification of deep caries, root remnants, and stage III periodontal disease. Maxillofacial MSCT performed by staging in patients with OPC should always be analysed in the detection of AP lesions, thereby helping to prevent delays in appropriate diagnosis and treatment. Pre-RT PAN should be carried out only in selected cases.

**Author Contributions:** Conceptualization, C.B., I.D. and C.N.; methodology, M.P., D.M. and V.R.; validation, L.G.L. and V.G.; investigation, C.B. and V.R.; resources, M.P. and C.N.; data curation, D.M. and L.F.; writing—original draft preparation, C.B., V.G. and C.N.; writing—review and editing, I.D., L.F. and C.N.; visualization, M.P. and L.G.L.; supervision, I.D., L.F. and C.N.; project administration, V.R. and C.N. All authors have read and agreed to the published version of the manuscript.

**Funding:** This research received no external funding.

**Institutional Review Board Statement:** The study was conducted according to the guidelines of the Declaration of Helsinki. This is a monocentric retrospective comparative study approved by the Ethical Review Board of the AOU Careggi (# 14122).

**Informed Consent Statement:** Written informed consent was obtained from all patients involved in the study.

**Data Availability Statement:** Not applicable.

**Conflicts of Interest:** The authors declare no conflict of interest.

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
