# Peer review of "Is Panoramic Radiography Really a Key Examination before Chemo-Radiotherapy Treatment for Oropharyngeal Cancer?"

_applsci, doi:10.3390/app11177965_

Round 1
Reviewer 1 Report
The title needs more data
Many methodological biases exist
Discussion Section is required more relevant data

Reviewer 2 Report
The article "Is panoramic radiography really a key examination before chemo-radiotherapy treatment for oropharyngeal cancer?" is very interesting for clinical practice.
- Abstract: the conclusion proposed by the authors is the expression of the results. It would be worth to write a good conclusion.
- Introduction: The advantages and disadvantages of panoramic radiography and MSCT for the diagnosis of the lesions to be analyzed in the study could be mentioned in this section.
- Discusion: in relation to the waiting time to perform the extraction of the teeth before the radiation treatment, which the authors suggest (10 days), what do the authors think about of the 14-day withdrawal period proposed by Irie et al,
Milena-Suemi Irie,1 Eduardo-Moura Mendes,2 Juliana-Simeão Borges,3 Luis-Gustavo-Gonzalez Osuna,2 Gustavo-Davi Rabelo,4 and Priscilla-Barbosa-Ferreira Soares 5 Periodontal therapy for patients before and after radiotherapy: A review of the literature and topics of interest for clinicians. Med Oral Patol Oral Cir Bucal. 2018; 23(5): e524–e530.
- Conclusions: What criteria would you use to consider suspected apical periodontitis to perform maxillofacial MSCT?
- In which cases would you recommend doing Pre-RTPAN?
- References:
- Are the 5 citations from Nardi et al. necessary?, specially number 26?
- Adapt the references to the norms of the journals.
Thank you very much
Reviewer 3 Report
The topic of the manuscript is the evaluation of the diagnostic accuracy of panoramic radiography in detecting oral infectious foci in OPC patients before radiotherapy, using MSCT imaging as the reference standard.
The title and the abstract of the article are informative. The Introduction briefly presents the problem of pre-irradiation dental-periodontal screening in patients with oropharyngeal carcinoma. The section “Material and Methods” precisely describes the study design, including a flowchart of selection criteria, as well as inclusion and exclusion criteria. The statistical analysis and the presentation of results are correctly performed. The Discussion is well-written, including the study limitations and the appropriate references from recent years. The Conclusions are good “take-home” messages.
In my opinion, the article in its current form can be published; alternatively, it is suggested to add references from 2021 (if possible). Congratulations to the authors for preparing the high-quality and clinically relevant manuscript.
